# Association between fine particulate matter (PM$_{2.5}$) and violence cases in South Korea: A nationwide time-stratified care-crossover study

Jiwoo Park[1‡], Jieun Oh[2‡], Hyewon Yoon[3], Ayoung Kim[2], Cinoo Kang[2], Dohoon Kwon[2], Jinah Park[2], Ho Kim[2], Whanhee Lee[4,5]*

1 Department of Information Convergence Engineering, Pusan National University, Yangsan, South Korea, 2 Department of Public Health Sciences, Graduate School of Public Health, Seoul National University, Seoul, South Korea, 3 Graduate School of Data Science, Pusan National University, Pusan, South Korea, 4 School of Biomedical Convergence Engineering, Pusan National University, Yangsan, South Korea, 5 Research and Management Center for Health Risk of Particulate Matter, Seoul, Republic of Korea

‡ JP and JO are contributed equally to this work as the co-first authors.
* whanhee.lee@pusan.ac.kr

**Data Availability Statement:** All files related to the Korea National Hospital Discharge In-depth Injury Survey are publicly available from the Korea National Hospital Discharge In-depth Injury Survey

## Abstract

Several studies reported the roles of short-term exposure to fine particulate matter (PM$_{2.5}$) on violent behaviors; however, existing findings had a limitation in assessing the population-representative association between violence and PM$_{2.5}$ due to the limited data availability: most studies have been based on homicides in monitored urban areas. This study collected violence data from the National Hospital Discharge In-depth Injury Survey in South Korea (2015–2019), based on population-representative samples. To cover unmonitored areas, we used the daily modeled PM$_{2.5}$, the predicted result driven by a machine-learning ensemble model covering all inland districts in South Korea (R$^2$>0.94). We evaluated the national association between short-term exposure to PM$_{2.5}$ and violence cases with a time-stratified case-crossover design. A total of 2,867 violence cases were included. We found an approximately linear association between short-term exposure to PM$_{2.5}$ (lag 0–2 days) and an increased risk of violence, with an estimated odd ratio (OR) per 10 μg/m$^3$ of PM$_{2.5}$ of 1.07 with 95% CI: 1.02–1.12. This relationship was more prominent in males and individuals aged 64 years or less than in females and individuals aged 65 years or older for the most part. The estimated excess fraction of violence cases attributable to PM$_{2.5}$ was 14.53% (95% CI: 4.54%–22.92%), and 6.42% (95% CI: 1.97%–10.26%) of the excess violence was attributable to non-compliance with the WHO guidelines (daily PM$_{2.5}$ > 15 μg/m$^3$). Our findings might be evidence of the need to establish elaborate action plans and stricter air quality guidelines to reduce the hazardous impacts of PM$_{2.5}$ on violence in South Korea.

database (https://www.kdca.go.kr/injury/biz/injury/recsroom/rawDta/rawDtaDwldMain.do#) for research purpose. People who want to download this data should submit the application form with their research objects and receive approval from the Korea Disease Control and Prevention Agency to use it.

**Funding:** WL was supported by the National Institute of Environmental Research (NIER) funded by the Ministry of Environment (MOE) of the Republic of Korea (NIER-2021-03-03-007). JP was supported by Institute of Information & communications Technology Planning & Evaluation (IITP) under the Artificial Intelligence Convergence Innovation Human Resources Development (IITP-2024-RS-2023-00254177) grant funded by the Korea government (MSIT). The funders had no role in study design, data collection and analysis, decision to publish, or preparation of the manuscript.

## Introduction

Mental disorders have long been recognized as one of the major social and public health problems, especially related to the COVID-19 pandemic [1]. In particular, among mental disorders, violence–which is one of the major intentional injuries–is a crucial public health concern. According to the WHO fact sheet regarding injuries and violence (as of 19 June 2024) [2], injuries and violence are responsible for an estimated 10% of all years of lives with disabilities, and violence-related injuries kill 1.25 million people every year in 2019 with 6.2 deaths per 100,000 persons died due to homicide globally. In South Korea, the homicide-related mortality rate was 0.8 per 100,000 in 2019 [3], and it was higher than in neighboring East Asian countries (0.2 and 0.3 per 100,000 in Japan and Singapore) [3]. Moreover, severe violence cases have been seriously and widely addressed in mass and social media because it is a critical risk factor that destroys human well-being and quality of life [4]. Therefore studies on violence are timely and important.

Numerous studies have examined the risk factors for violence including socioeconomic and individual factors, and even genetics [5, 6]. Concurrently, rich studies have consistently reported that particulate matter (PM) is one of the important environmental risk factors for unintentional injuries including violence [7]. Historically, the hypothesis that exposure to PM might increase neurological disorders associated with cognitive responses, impulsiveness, and depression has been addressed in epidemiological studies [8, 9] as well as in laboratory studies [10, 11].

Nonetheless, previous studies on PM and violence have several limitations. First, most of the previous studies investigating this topic used homicide data [12, 13], thus there could be knowledge gaps regarding the impacts of PM on mild or moderate violence cases. Second, due to the limited exposure or violence data, many studies have included selected areas with air pollution monitoring stations with a sufficient sample size for statistical analyses [13, 14]–mostly metropolitan or urban areas could satisfy these conditions–therefore, there could be selection biases regarding the limited areas, especially if the nationwide association should be assessed for the population-representative public health policy. Lastly, most studies on air pollution and violence or crime have been conducted in the United States [9, 15], and studies in Asian regions with different lifestyles, diets, genes, social disparities, and demographic characteristics are scarce. Especially, South Korea is one of the notable countries related to mental health, because they showed a suicide rate of 24.1 per 100,000 people in 2020, which is the highest rate in the OECD countries [16]. Thus, studies on air pollution and violence in South Korea could benefit their populations and other Asian countries with similar environments, cultures, and public health issues.

Therefore, to address these gaps in knowledge, this study aims to assess the nationwide risks of fine particulate matter (PM$_{2.5}$) on all types of violence cases, which allows us to provide wider information on violence than that from the homicide data that could only cover severe cases. This study used the national population-representative survey data covering all districts in South Korea, provided by the Korea Disease Control and Prevention Agency (2015–2019). Further, we used a nationwide machine learning-based ensemble prediction model for daily PM$_{2.5}$ with excellent spatial resolution (1 km$^2$) and accuracy (R$^2$>0.94) to evaluate the less-biased association between short-term exposure to particulate matter and violence.

## Materials and methods

### Ethical approval

Not requested. This study used secondary and publicly available data. This data did not include any information related to the personal identification.

## Data on violence

We obtained national data on hospital visits due to violence from 2015 to 2019 across 247 districts in Korea. Specifically, we collected the Korea National Hospital Discharge In-depth Injury Survey, officially operated annually and provided by the Korea Disease Control and Prevention Agency to investigate the national status regarding all types of violence and generate relevant statistics.

This survey annually sampled 150,000 to 300,000 (it has increased over the years) people discharged from the general hospital, and the target population is all people discharged from the general hospital residing in South Korea from 2005. Also, to get population representativeness, the survey adopted the two-stage stratified-cluster systematic sampling method, based on regions, age-sex structures, and the number of hospital beds in the selected hospitals. Theoretically, the entire general hospital should be a target for the survey; however, due to practical reasons, the survey targeted general hospitals with 100 beds or more. Among the general hospitals with 100 beds or more, the survey sampled a total of 250 hospitals using the Neyman allocation method [17] based on the number of hospital beds in each hospital.

This survey investigated all types of injuries (both intentional and unintentional) using the related electronic medical records from the selected hospitals. The injury was defined as S00-T98 code (certain other consequences by injury, addiction, and externalities) in main diagnosis or sub-diagnosis by KCD-8th and ICD-9-CM Vol.Ⅲ. Each recorded case also includes information on the patient's sex, age, residential address (district; *"si/gun/gu"* in Korean), date of hospitalization, mechanism, etc. From this nationwide survey, this study collected violent cases based on medical information on the intentionality of the injury (intentional/unintentional) and types of intentional injuries (violence/suicide/poisoning, etc.) which included KCD-8th X85-Y09 code patients. This survey clarifies violence as violence between people, such as being punched by a person, beaten with a blunt instrument, or raped, except for violence under legal adjustment.

## Modeled PM$_{2.5}$ and environmental data

To cover unmonitored districts, a nationwide daily modeled PM$_{2.5}$ (the predicted value using a machine-learning ensemble prediction model with a 1 km$^2$ spatial resolution) was used as an exposure for all districts. The modeled PM$_{2.5}$ (24-hour average) was provided by the AiMS-CREATE team, and their prediction models were used in previous studies [18, 19]. The ensemble model incorporates three machine-learning algorithms (random forest, extreme gradient boosting, and deep neural network). Detailed information on the model is reported in the Supplementary Materials: "1. Air pollution prediction models" and S1 Table in S1 File. The prediction models for PM$_{2.5}$ showed excellent performances: across districts, a cross-validated R$^2$ of 0.944. Daily concentration predictions at 1 km$^2$ were aggregated to each district by averaging the predictions at grid cells with centroid points inside the boundary of each district. S1-S3 Tables in S1 File displays the prediction performance of the PM$_{2.5}$ ensemble prediction model used in this study.

We collected meteorological variables as time-varying confounders from the ERA-5 Land global reanalysis dataset [20], and these variables include 24-hour average 2-m air temperature (K), relative humidity (%), and precipitation (m). This ERA-5 dataset has a horizontal resolution of 0.1˚ x 0.1˚, with a native spatial resolution of 9–11 km, and we aggregated it to district unit ("Si/Gun/Gu") by averaging the values at grid cells with centroid points inside the boundary of that district.

## Study design

This study adopted a time-stratified case-crossover design to estimate the association between short-term PM$_{2.5}$ (lag 0–2) and violence. We defined a case day as the date of each outcome and matched control days as days with the day of the week within the same month in the same year. This time-stratified self-matching controlled for confounding variables that do not change substantially in a month, such as age, sex, weight, diet, and other individual-level time invariant health behavior characteristics, and also district-level regional variables like population density, gross regional domestic product, and other socio-environmental factors including park and medical accessibilities, population composition, and access to grocery shops [21]. Furthermore, the time-stratified matching controlled potential confounding that varies across weekdays and weekends, with bidirectional control day selection that can remove biases from seasonality and long-term trends of PM$_{2.5}$ and outcomes [22]. Therefore, the time-stratified case-crossover design has been widely used in studies evaluating the risk of short-term environmental exposure on acute health outcomes [21, 23–26].

## Statistical analysis

We estimated the risk of violence associated with short-term exposure to PM$_{2.5}$ using a conditional logistic regression model. For the main model, we selected a mean value of lag 0 to lag 2 PM$_{2.5}$ exposure to address the average health risks associated with the same and the previous days' exposures based on existing relevant studies [21, 27, 28]. We adjusted indicator variables of holiday and daily temperatures. To control potentially nonlinear confounding, temperatures were controlled using a cross-basis function with a natural cubic spline with four degrees of freedom for an exposure-response relationship, and a natural cubic spline with intercepts and one internal knot (at lag 1) for a lag-response relationship over the lag of two days. The relationship between relatively short-term temperature (one or two lag days) and violence, suicide, or other acute mental disorders has been identified in many related epidemiological studies [13, 29]. We calculated the odds ratio (OR) for a 10 µg/m$^3$ increase in PM$_{2.5}$.

## Subgroup analysis

Subgroup analyses (sex, age groups, Urban/Rural, GRDP (Gross Regional Domestic Product) High/Low, and seasons) were also conducted to identify the high-risk populations. Sex groups were divided into two categories: Male and Female. Age groups were also distinguished into two categories: people aged 65 years or older and those aged less than 64 years. The region was evaluated in largely two parts: 1) urban and rural and 2) high and low areas based on GRDP (Gross Regional Domestic Product) per 100,000 persons. First, we classified all study districts (si/gun/gu) into urban (si and gu) and rural (gun) districts, based on the Local Autonomy Act of Korea, which has been used in previous studies [30, 31]. We collected annual district-level GRDP per 100,000 people from 2015 to 2019 from Statistics Korea. Then, we calculated the district-specific average GRDP per 100,000 people during the study period. We divided study districts into two categories (high/low) based on the median value of average GRDP per 100,000 persons. Two seasons were also considered: the warm season (April–September) and the cold season (October–March). To evaluate the different associations between PM$_{2.5}$ and violence for each subgroup, we additionally created a case-crossover dataset for each sex, age group, urban/rural, GRDP high/low group, and season and repeated the main analysis. Further, we conducted the Wald test based on the independent assumption to check whether OR estimates between subgroups are statistically different (H$_0$: there is no difference).

## Excess violence burden attributable to PM$_{2.5}$

Odd ratio estimates for short-term exposure to PM$_{2.5}$ were translated into excess violence cases attributable to PM$_{2.5}$, to demonstrate the excess burden attributable to PM$_{2.5}$ exposures during the study period. The estimates of the excess burden (either as an attributable burden or relative excess measures of outcomes) are widely used to assess the actual and quantitative public health impacts of extreme exposures (e.g., heatwaves or high-level air pollution).

To calculate it, we created district-specific time-series datasets including daily mean PM$_{2.5}$ and daily counts of violence. For each time-series data, we calculated the daily excess violence cases attributable to PM$_{2.5}$ using the corresponding estimated ORs for the PM$_{2.5}$ concentrations of each day (daily excess violence cases attributable to daily PM$_{2.5}$:
sum of violecne cases$_t*[OR_t - 1]/OR_t$; where $t$ indicates the day, and OR$_t$ presents the daily OR for the PM$_{2.5}$ concentrations of day $t$). Here in, the OR estimates were used as the approximated values of relative risk (RR) that were originally used to calculate the attributable risk estimates. When the proportion of the outcome onset is substantially small (in general, 20% or less), the OR estimates can be regarded as an approximated estimate of the RR estimates [32]. In this study, the total number of outcomes (violence cases) was 2,867 cases during the find years (2015 to 2019), and when the total number of investigation cases of the Korea National Hospital Discharge In-depth Injury Survey (sampled 150,000 to 300,000 per year). Therefore, the probability of our outcome was substantially small. Thus, we used OR estimates as the approximated values of RR estimates in this study.

The sum of daily excess violence cases indicates the total excess violence cases attributable to PM$_{2.5}$ during the study period, and its ratio with the total number of violence provides the total excess fraction (%) of violence cases attributable to short-term PM$_{2.5}$ exposures. Since a lot of previous studies reported that the air pollution impacts on health persist at low levels [28], we set a minimum PM$_{2.5}$ concentration during the study period as the reference to calculate excess violence cases attributable to PM$_{2.5}$ (i.e., the whole range of PM$_{2.5}$), although we recognize that this includes natural background concentrations.

Further, we assessed the excess violence burden attributable to non-compliance with the current guidelines regarding PM$_{2.5}$: the WHO guideline, the National Ambient Air Quality Standards (NAAQS) in the United States, and the Korean Air Quality Standards (KAS) in South Korea. These guidelines have been established to provide quantitative recommendations for air quality management, and exceeding these levels is associated with important risks to public health. Therefore, we calculated excess violence cases only for the subset of days with PM$_{2.5}$ levels above (i.e. non-compliance) the WHO air quality guidelines (daily average PM$_{2.5}$ $\leq$ 15 μg/m$^3$), and the NAAQS and KAS (daily average PM$_{2.5}$ $\leq$ 35 μg/m$^3$), respectively, to illustrate the burden that could be prevented or reduced through compliance with the guidelines. The Monte Carlo simulations were used to compute the confidence intervals of each estimate, with 1,000 replicates [26, 33].

## Sensitivity analysis

We performed a series of sensitivity analyses to examine whether our results were consistent with additional different modeling specifications in the total population. Sensitivity analyses were carried out about two aspects. First, we tried to show the robustness of PM$_{2.5}$'s lag period. Single lag from 0 to 3 was executed and the moving average from lag 0–1 to 0–3 was tested. Second, for daily temperature, which was used as a covariate, we applied 3 and 7 days of lag period, and 3 and 5 degrees of freedom for an exposure-response relationship.

## Results

Table 1 presents the descriptive statistics on violence outcomes during the study period (2015–2019). Total counts of violence were 2,867, of which 67.6% were males and 92.8% occurred in individuals aged less than 65 years. Annually, approximately 544 to 612 cases of violence were reported in South Korea (S4 Table in S1 File). Collisions accounted for the highest % of injury mechanisms at 88.7% (S5 Table in S1 File). Complete and partial recovery was the most common treatment outcome in 96.16% of cases (S6 Table in S1 File). Fig 1 shows geographical distributions of average PM$_{2.5}$ during the study period. The national annual average concentration of PM$_{2.5}$ was 23.71 μg/m$^3$, with the lowest level recorded in 2018 at 21.71 μg/m$^3$ and the highest in 2015 at 25.74 μg/m$^3$ (S1 Table in S1 File).

Fig 2 presents the nonlinear exposure-response curve between PM$_{2.5}$ and violence. We found the association between PM$_{2.5}$ and violence cases was approximately linearly positive; thus, we presented risk estimates based on a linear association after Fig 2.

Fig 3 shows the association between PM$_{2.5}$ and violence in the total population and by subgroup. In the total population, the association between PM$_{2.5}$ and violence was evident with an OR of 1.07 with 95% CI: 1.02–1.12. The association was weakly higher in rural areas (OR: 1.23, 95% CI: 1.02–1.48) than in urban areas (OR: 1.06, 95% CI: 1.01–1.12) (Wald test p-value: 0.096). Further, the association between PM$_{2.5}$ and violence was statistically pronounced in cold seasons (OR: 1.08, 95% CI: 1.02–1.14) compared to warm seasons (1.05, 0.96–1.15), and the association was more pronounced in people aged 65 or older (OR: 1.33, 95% CI: 1.04–1.71) than those aged 64 or less (OR: 1.07, 95% CI: 1.00–1.13) in cold seasons (Wald test p-value: 0.042). This pattern remained consistent even when age groups were further subdivided (S7 Table in S1 File).

Table 2 reports the excess fractions of violence attributable to PM$_{2.5}$. For the total population, about 14.53% (4.54–22.92) of violence cases could be attributable to PM$_{2.5}$ exposures. Approximately 6.42% (1.97–10.26) and 1.11% (0.34–1.77) of excess violence cases were

**Table 1. Descriptive information on violence cases among Korea National Hospital Discharge In-depth Injury Survey data during the study period (2015–2019) in South Korea.** Holidays include Official public holidays in Korea. We classified the study districts (si/gun/gu) into urban (si and gu) and rural (gun) districts. GRDP High and Low areas are separated by a median value.

|  |  | N | % |
|---|---|---|---|
| **Total** |  | 2867 | 100 |
| **Sex** | **Males** | 1937 | 67.6 |
|  | **Females** | 930 | 32.4 |
| **Age** | **0–64 years** | 2660 | 92.8 |
|  | 0–19 years | 442 | 15.4 |
|  | 20–39 years | 1023 | 35.7 |
|  | 40–64 years | 1195 | 41.7 |
|  | **65+ years** | 207 | 7.2 |
|  | 65–79 years | 174 | 6.1 |
|  | 80+ years | 33 | 1.2 |
| **Holiday** | **Yes** | 106 | 3.7 |
|  | **No** | 2761 | 96.3 |
| **Urban/Rural** | **Urban** | 2568 | 89.6 |
|  | **Rural** | 299 | 10.4 |
| **GRDP per person** | **High** | 1311 | 45.7 |
|  | **Low** | 1556 | 54.3 |

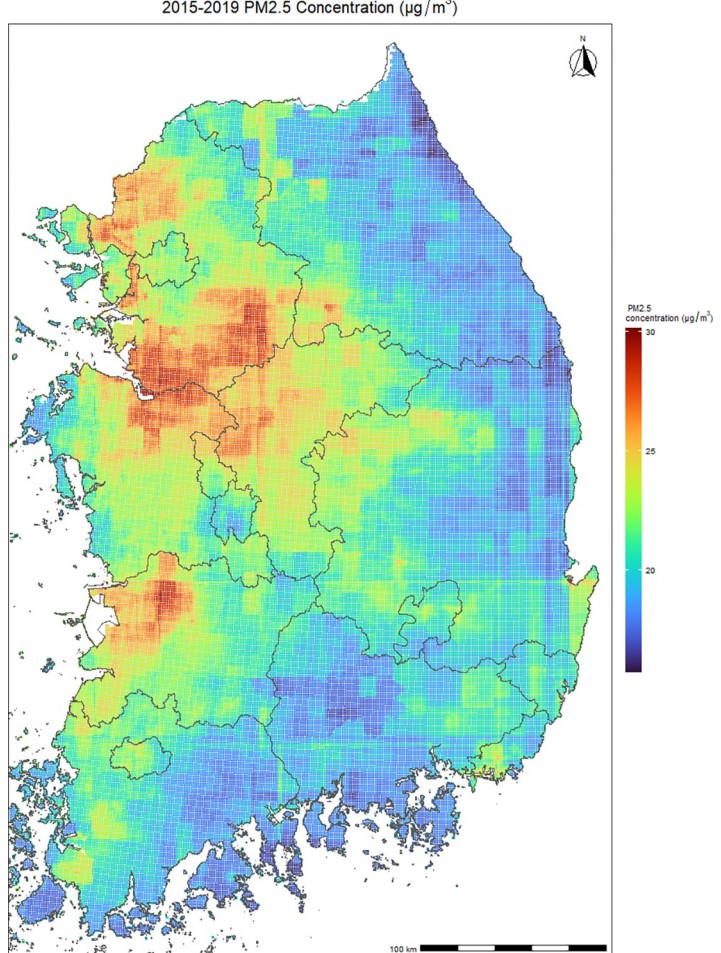

**Fig 1. Geographical distributions of the annual averages of daily average PM$_{2.5}$ (fine particulate matter; μg/m$^3$) in South Korea from 2015 through 2019 using a machine-learning ensemble prediction model with a 1 km$^2$ spatial resolution.** The prediction models got a 0.944 R$^2$ score across districts. Daily concentration predictions at 1 km$^2$ were aggregated in each district.

attributable to non-compliance with the WHO air quality guidelines (15 μg/m$^3$) and NAAQS and KAS (35 μg/m$^3$), respectively. The excess burden was also higher in males and individuals aged 64 years or less than in females and those aged 65 years or older. The burden of violence related to PM$_{2.5}$ was more pronounced during the cold season compared to the warm season in the total population and all subgroups.

Lastly, the results of sensitivity analyses showed that our main results were consistent with different modeling selections (S8 Table in S1 File).

## Discussion

To our knowledge, this is the first and largest epidemiological study investigating the national association between short-term PM$_{2.5}$ exposure and violence with a population-representative survey dataset in South Korea. The association between short-term PM$_{2.5}$ and violence was statistically evident in the total population, and the association was more pronounced in males and individuals aged 64 years or less, compared to females and individuals aged 65 years or

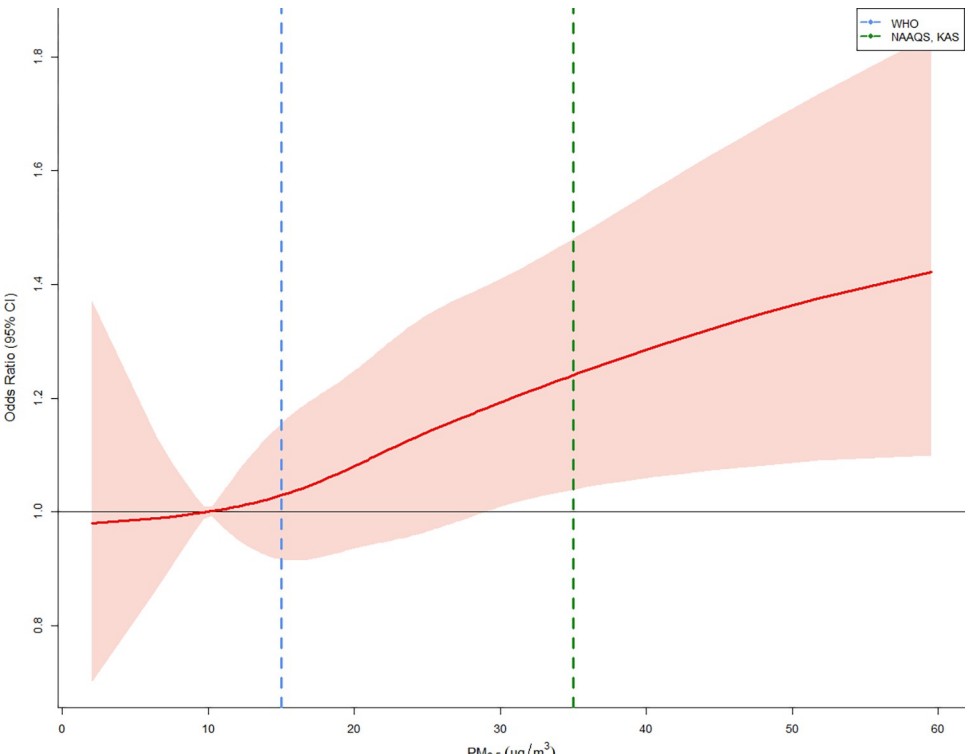

**Fig 2. Flexible association between short-term exposure to PM$_{2.5}$ and violence cases.** Each Blue and Green dashed lines indicate WHO: World Health Organization air quality guidelines (daily average PM$_{2.5}$: 15 μg/m$^3$), NAAQS: National Ambient Air Quality Standards (daily average PM$_{2.5}$: 35 μg/m$^3$) in the United States, and KAS: Korean Air Quality Standards (daily average PM$_{2.5}$: 35 μg/m$^3$), respectively. A conditional logistic regression model, adjusted for holiday and daily mean temperatures, within a time-stratified case-crossover design, was used to estimate odds ratios and 95% confidence intervals.

older. We also found that the excess fractions of violence attributable to non-compliance with WHO guidelines and NAAQS and KAS were 6.42% and 1.11%, respectively.

It is not long since violence was addressed from a public health perspective [34]. Throughout the 1970s and 1980s, the risk of suicide and homicide steadily increased in vulnerable populations, and this trend prompted responses from governments. In particular, intentional injuries such as violence are viewed as preventable [35], highlighting the importance of identifying modifiable risk factors. These include demographic factors (i.e., age, sex, and ethnicity), socioeconomic factors (i.e., education and employment status), and environmental factors (i.e., neighborhood safety and exposure to pollution). Furthermore, recent studies have reported consistent evidence that environmental factors might be associated with violence. Previous studies suggested that sudden weather changes (rising temperature, rain, an increase in humidity) can increase violent crimes, such as homicides, assaults, and sex offenses [36], and some theories (thermal or climatic discomfort can increase impulsivity and aggression, and changes in routine activities due to changing weather, such as outdoor events and social engagements) supports these results on weather and violence [36, 37]. In this context, the findings of this study are also anticipated to contribute valuable evidence for developing environmental epidemiological strategies aimed at violence prevention.

There has been a growing literature that has examined the association between air pollution and assault or homicides [9, 13, 38]. One study in the US found that each 10 μg/m$^3$ change in daily PM$_{2.5}$ was associated with a 1.17% increase in violent crime rates, with a particularly

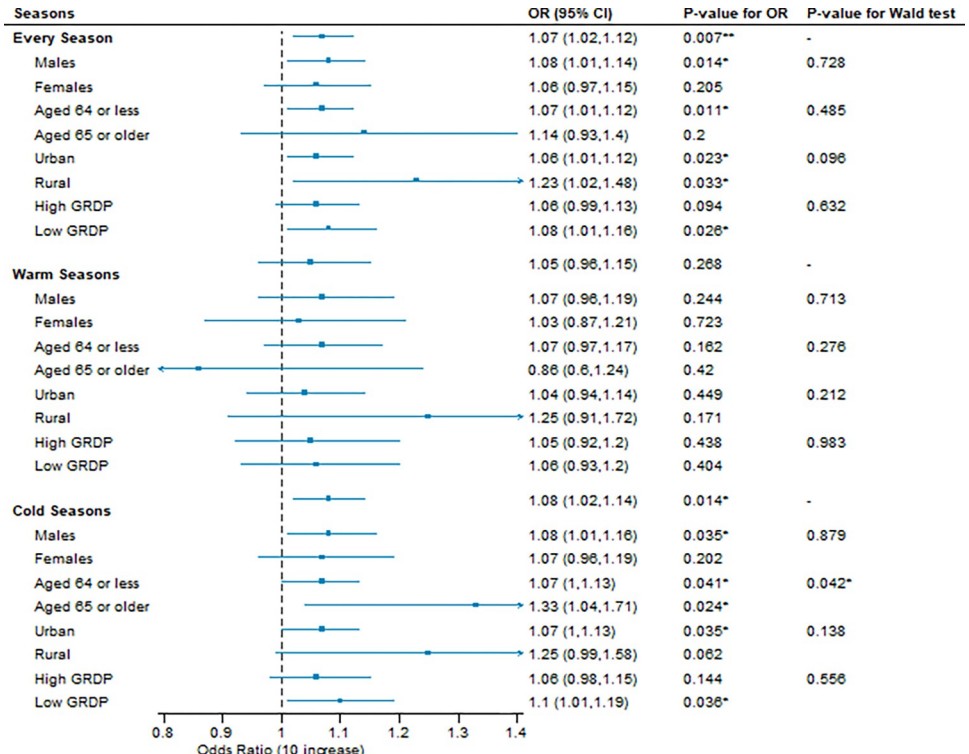

**Fig 3. Associations between short-term exposure to PM$_{2.5}$ (lag 0–2) and violence cases by season and subgroup.**
OR: Odds ratio per 10 μg/m$^3$ of PM$_{2.5}$. A conditional logistic regression model, adjusted for holiday and daily mean temperatures, within a time-stratified case-crossover design, was used to estimate odds ratios and 95% confidence intervals. Wald type test was used to test the effect modification. We classified the study districts (si/gun/gu) into urban (si and gu) and rural (gun) districts. GRDP High and Low areas are separated by a median value.

strong association observed in assault [9]. Similarly, a study in Seoul, South Korea, found that exposure to air pollutants is related to a risk of injuries, with a 1.08% increase in assault injuries for each interquartile range increase of PM$_{2.5}$ [38]. On the other hand, one study in California, US, on suicides and homicides found a significant association with ambient temperatures, but not with air pollutants [13]. Likewise, there have been no consistent findings on the association between air pollution and violence, and considering that most of the previous studies have been conducted in specific cities or regions, this nationwide study is expected to contribute to the limited literature.

Some plausible hypotheses have been suggested to explain the link between increased PM$_{2.5}$ and the risk of violence. Repeated exposure to PM$_{2.5}$ can cause immune dysregulation and trigger inflammatory processes, increasing the possibility of negative responses to various stressors [39]. The entry of PM$_{2.5}$ into the brain can diminish cognitive resources needed to guide choices and behaviors, potentially leading to cognitive decline [40]. These damages to the neurologic system have been shown to affect judgment and emotional processing, which can cause mental disorders such as depression and aggressive behavior [8, 41]. Although we could not clearly distinguish between victims or perpetrators of violence due to data limitations, it can be inferred that exposure to PM$_{2.5}$ may affect violence and related mental health problems, increasing the risk of being involved in violent incidents.

We found a greater association between short-term increases in PM$_{2.5}$ and violence in males than in females. Globally, males are overwhelmingly more likely to be involved in violent crimes, both as perpetrators and victims [42], as shown in the dataset of this study. Some

**Table 2. Excess numbers and fractions of violence cases attributable to PM$_{2.5}$ and non-compliance with the current WHO air quality guidelines, NAAQS in the United States, and KAS in South Korea.** WHO: World Health Organization, NAAQS: National Ambient Air Quality Standards. KAS: Korean Air Quality Standards. We classified the study districts (si/gun/gu) into urban (si and gu) and rural (gun) districts. GRDP High and Low areas are separated by a median value.

| | | | Excess numbers (%) | | Excess fractions (%) | |
|---|---|---|---|---|---|---|
| **Total season** | **Total population** | Whole PM$_{2.5}$ | 416.58 | (130.21, 657.14) | 14.53 | (4.54, 22.92) |
| | | WHO guidelines (>15 μg/m$^3$) | 184.08 | (56.38, 294.19) | 6.42 | (1.97, 10.26) |
| | | NAAQS & KAS (> 35 μg/m$^3$) | 31.77 | (9.79, 50.62) | 1.11 | (0.34, 1.77) |
| | **Males** | Whole PM$_{2.5}$ | 299.36 | (47.45, 508.15) | 15.45 | (2.45, 26.23) |
| | | WHO guidelines (>15 μg/m$^3$) | 133.77 | (20.66, 230.69) | 6.91 | (1.07, 11.91) |
| | | NAAQS & KAS (> 35 μg/m$^3$) | 24.78 | (3.86, 42.61) | 1.28 | (0.2, 2.2) |
| | **Females** | Whole PM$_{2.5}$ | 109.89 | (-73.94, 263.35) | 11.82 | (-7.95, 28.32) |
| | | WHO guidelines (>15 μg/m$^3$) | 48.00 | (-30.78, 117.54) | 5.16 | (-3.31, 12.64) |
| | | NAAQS & KAS (> 35 μg/m$^3$) | 6.95 | (-4.54, 16.94) | 0.75 | (-0.49, 1.82) |
| | **0–64 years** | Whole PM$_{2.5}$ | 372.77 | (96.33, 601.02) | 14.01 | (3.62, 22.59) |
| | | WHO guidelines (>15 μg/m$^3$) | 164.74 | (41.68, 269.15) | 6.19 | (1.57, 10.12) |
| | | NAAQS & KAS (> 35 μg/m$^3$) | 28.39 | (7.23, 46.24) | 1.07 | (0.27, 1.74) |
| | **65 years or older** | Whole PM$_{2.5}$ | 47.34 | (-49.18, 101.25) | 22.87 | (-23.76, 48.91) |
| | | WHO guidelines (>15 μg/m$^3$) | 21.53 | (-20.27, 47.8) | 10.4 | (-9.79, 23.09) |
| | | NAAQS & KAS (> 35 μg/m$^3$) | 3.79 | (-3.75, 8.5) | 1.83 | (-1.81, 4.1) |
| | **Urban** | Whole PM$_{2.5}$ | 320.17 | (62.91, 581.01) | 12.47 | (2.45, 22.63) |
| | | WHO guidelines (>15 μg/m$^3$) | 142.23 | (27.37, 261.99) | 5.54 | (1.07, 10.2) |
| | | NAAQS & KAS (> 35 μg/m$^3$) | 25.26 | (4.9, 46.33) | 0.98 | (0.19, 1.8) |
| | **Rural** | Whole PM$_{2.5}$ | 103.97 | (10.83, 166.89) | 34.77 | (3.62, 55.82) |
| | | WHO guidelines (>15 μg/m$^3$) | 45.48 | (4.35, 76.58) | 15.21 | (1.45, 25.61) |
| | | NAAQS & KAS (> 35 μg/m$^3$) | 5.89 | (0.55, 10.04) | 1.97 | (0.18, 3.36) |
| | **GRDP High** | Whole PM$_{2.5}$ | 173.77 | (-13.89, 337.93) | 13.25 | (-2.37, 25.78) |
| | | WHO guidelines (>15 μg/m$^3$) | 80.07 | (-13.89, 158.09) | 6.11 | (-1.06, 12.06) |
| | | NAAQS & KAS (> 35 μg/m$^3$) | 15.91 | (-2.79, 31.3) | 1.21 | (-0.21, 2.39) |
| | **GRDP Low** | Whole PM$_{2.5}$ | 237.09 | (19.08, 419.70) | 15.24 | (1.23, 26.97) |
| | | WHO guidelines (>15 μg/m$^3$) | 101.05 | (7.86, 182.56) | 6.49 | (0.51, 11.73) |
| | | NAAQS & KAS (> 35 μg/m$^3$) | 14.67 | (1.15, 26.44) | 0.94 | (0.07, 1.70) |
| **Warm seasons** | **Total population** | Whole PM$_{2.5}$ | 139.75 | (-147.1, 368.71) | 9.02 | (-9.50, 23.8) |
| | | WHO guidelines (>15 μg/m$^3$) | 51.49 | (-50.94, 138.47) | 3.32 | (-3.29, 8.94) |
| | | NAAQS & KAS (> 35 μg/m$^3$) | 3.63 | (-3.48, 9.84) | 0.23 | (-0.22, 0.64) |
| | **Males** | Whole PM$_{2.5}$ | 123.20 | (-92.84, 297.97) | 11.73 | (-8.84, 28.38) |
| | | WHO guidelines (>15 μg/m$^3$) | 45.12 | (-31.82, 111.92) | 4.3 | (-3.03, 10.66) |
| | | NAAQS & KAS (> 35 μg/m$^3$) | 3.32 | (-2.29, 8.32) | 0.32 | (-0.22, 0.79) |
| | **Females** | Whole PM$_{2.5}$ | 25.46 | (-146.19, 160.04) | 5.10 | (-29.3, 32.07) |
| | | WHO guidelines (>15 μg/m$^3$) | 10.55 | (-49.79, 63.25) | 2.11 | (-9.98, 12.67) |
| | | NAAQS & KAS (> 35 μg/m$^3$) | 0.72 | (-2.99, 4.22) | 0.14 | (-0.6, 0.85) |
| | **0–64 years** | Whole PM$_{2.5}$ | 176.21 | (-72.62, 398.96) | 12.28 | (-5.06, 27.8) |
| | | WHO guidelines (>15 μg/m$^3$) | 64.94 | (-25.31, 151.04) | 4.53 | (-1.76, 10.53) |
| | | NAAQS & KAS (> 35 μg/m$^3$) | 4.60 | (-1.75, 10.82) | 0.32 | (-0.12, 0.75) |
| | **65 years or older** | Whole PM$_{2.5}$ | -57.79 | (-249.01, 35.07) | -50.7 | (-218.43, 30.76) |
| | | WHO guidelines (>15 μg/m$^3$) | -17.19 | (-67.45, 14.07) | -15.08 | (-59.16, 12.35) |
| | | NAAQS & KAS (> 35 μg/m$^3$) | -0.95 | (-3.47, 0.99) | -0.84 | (-3.04, 0.87) |
| | **Urban** | Whole PM$_{2.5}$ | 86.13 | (-186.27, 305.19) | 6.22 | (-13.45, 22.04) |
| | | WHO guidelines (>15 μg/m$^3$) | 32.21 | (-64.47, 114.98) | 2.33 | (-4.65, 8.3) |
| | | NAAQS & KAS (> 35 μg/m$^3$) | 2.26 | (-4.34, 8.09) | 0.16 | (-0.31, 0.58) |
| | **Rural** | Whole PM$_{2.5}$ | 51.30 | (-28.18, 103.46) | 31.28 | (-17.18, 63.08) |

(*Continued*)

**Table 2.** (Continued)

| | | | Excess numbers (%) | | Excess fractions (%) | |
|---|---|---|---|---|---|---|
| | | WHO guidelines (>15 μg/m$^3$) | 19.27 | (-9.00, 41.67) | 11.75 | (-5.49, 25.41) |
| | | NAAQS & KAS (> 35 μg/m$^3$) | 1.47 | (-0.68, 3.19) | 0.89 | (-0.42, 1.95) |
| | GRDP High | Whole PM$_{2.5}$ | 68.28 | (-114.45, 212.04) | 9.84 | (-16.49, 30.55) |
| | | WHO guidelines (>15 μg/m$^3$) | 26.34 | (-40.42, 83.08) | 3.79 | (-5.82, 11.97) |
| | | NAAQS & KAS (> 35 μg/m$^3$) | 2.34 | (-3.46, 7.42) | 0.34 | (-0.50, 1.07) |
| | GRDP Low | Whole PM$_{2.5}$ | 75.04 | (-128.23, 241.67) | 8.78 | (-15.00, 28.27) |
| | | WHO guidelines (>15 μg/m$^3$) | 27.50 | (-43.54, 90.04) | 3.22 | (-4.98, 10.53) |
| | | NAAQS & KAS (> 35 μg/m$^3$) | 1.49 | (-2.16, 4.94) | 0.17 | (-0.25, 0.58) |
| Cold seasons | Total population | Whole PM$_{2.5}$ | 233.62 | (60.63, 380.68) | 17.73 | (4.6, 28.88) |
| | | WHO guidelines (>15 μg/m$^3$) | 119.33 | (30.22, 197.62) | 9.05 | (2.29, 14.99) |
| | | NAAQS & KAS (> 35 μg/m$^3$) | 28.41 | (7.2, 47.07) | 2.16 | (0.55, 3.57) |
| | Males | Whole PM$_{2.5}$ | 164.11 | (19.9, 292.52) | 18.50 | (2.24, 32.98) |
| | | WHO guidelines (>15 μg/m$^3$) | 85.55 | (10.06, 155.59) | 9.64 | (1.13, 17.54) |
| | | NAAQS & KAS (> 35 μg/m$^3$) | 21.66 | (2.54, 39.5) | 2.44 | (0.29, 4.45) |
| | Females | Whole PM$_{2.5}$ | 62.69 | (-46.26, 146.17) | 14.55 | (-10.73, 33.91) |
| | | WHO guidelines (>15 μg/m$^3$) | 31.05 | (-21.69, 73.97) | 7.2 | (-5.03, 17.16) |
| | | NAAQS & KAS (> 35 μg/m$^3$) | 6.30 | (-4.47, 14.96) | 1.46 | (-1.04, 3.47) |
| | 0–64 years | Whole PM$_{2.5}$ | 186.68 | (2.61, 342.28) | 15.24 | (0.21, 27.94) |
| | | WHO guidelines (>15 μg/m$^3$) | 95.4 | (1.3, 177.88) | 7.79 | (0.11, 14.52) |
| | | NAAQS & KAS (> 35 μg/m$^3$) | 22.57 | (0.31, 42.09) | 1.84 | (0.03, 3.44) |
| | 65 years or older | Whole PM$_{2.5}$ | 41.16 | (5.73, 60.99) | 44.25 | (6.16, 65.58) |
| | | WHO guidelines (>15 μg/m$^3$) | 21.41 | (2.76, 33.2) | 23.03 | (2.96, 35.70) |
| | | NAAQS & KAS (> 35 μg/m$^3$) | 5.72 | (0.72, 9.13) | 6.15 | (0.77, 9.81) |
| | Urban | Whole PM$_{2.5}$ | 187.08 | (27.78, 331.17) | 15.81 | (2.35, 27.99) |
| | | WHO guidelines (>15 μg/m$^3$) | 96.05 | (13.91, 172.8) | 8.12 | (1.18, 14.61) |
| | | NAAQS & KAS (> 35 μg/m$^3$) | 23.63 | (3.43, 42.52) | 2.00 | (0.29, 3.59) |
| | Rural | Whole PM$_{2.5}$ | 56.31 | (-1.59, 92.44) | 41.71 | (-1.18, 68.47) |
| | | WHO guidelines (>15 μg/m$^3$) | 29.11 | (-0.73, 51.03) | 21.56 | (-0.54, 37.8) |
| | | NAAQS & KAS (> 35 μg/m$^3$) | 4.89 | (-0.12, 8.84) | 3.62 | (-0.09, 6.55) |
| | GRDP High | Whole PM$_{2.5}$ | 96.11 | (-36.64, 204.64) | 15.58 | (-6.26, 33.17) |
| | | WHO guidelines (>15 μg/m$^3$) | 51.23 | (-19.81, 111.20) | 8.30 | (-3.21, 18.02) |
| | | NAAQS & KAS (> 35 μg/m$^3$) | 13.54 | (-5.25, 29.41) | 2.19 | (-0.85, 4.77) |
| | GRDP Low | Whole PM$_{2.5}$ | 137.2 | (13.83, 237.86) | 19.57 | (1.97, 33.93) |
| | | WHO guidelines (>15 μg/m$^3$) | 67.64 | (6.56, 120.01) | 9.65 | (0.94, 17.12) |
| | | NAAQS & KAS (> 35 μg/m$^3$) | 14.15 | (1.37, 25.17) | 2.02 | (0.20, 3.59) |

researchers have hypothesized that masculine norms, which emphasize that men should be strong and willing to take risks, are strongly linked to interpersonal violence [43], and that gender can shape individuals' activity patterns, such as labor proclivities, which may affect the magnitude of exposure to air pollution [44]. We also found a more evident association between PM$_{2.5}$ and violence among individuals aged 64 years or less, which is inconsistent with previous findings that air pollution has a greater impact on mortality and morbidity in the elderly [45]. However, this inconsistency may be due to different lifestyles and behaviors. In the case of air pollution and violence, younger people, who tend to be more socially active and spend more time outdoors, may be more likely to be affected by PM$_{2.5}$ and involved in violent situations. More research is needed to understand these possible sex and age differences in the effects of air pollution on violence.

Our finding of stronger effects of PM$_{2.5}$ in the cold season is consistent with several previous studies [14, 45]. One study in Shanghai, China, found that the impacts of air pollutants on mortality from all causes and cardiorespiratory diseases were more evident in the cold season than in warm seasons [45]. Another study in California, US, also found a stronger association between PM$_{2.5}$ and homicide during cold seasons [14]. In this study, cold seasons have been found to have higher concentrations of PM$_{2.5}$ compared to warm seasons in South Korea, but more research is needed to understand the seasonal variations in the chemical composition and toxicity of PM$_{2.5}$. Additionally, it is worth noting that in this study only those aged 65 years or older had a significantly higher risk during the cold season compared to all seasons, while the risk estimates for those aged 64 years or less did not change by season. Older adults, who may experience weakened immune systems, worsening of underlying conditions, and decline in physical function during the cold season, can be more vulnerable to exposure to air pollution and risk of injury during this season. Further, alcohol consumption, which is closely linked to violent behavior [46], tends to increase during the cold months; cold weather and reduced sunlight, as well as holidays such as New Year's and Christmas, are associated with higher alcohol intake [47–49]. This is especially concerning for older adults (65+ years), as increased alcohol consumption can worsen pre-existing vulnerabilities to physical and mental health problems [50]. Moreover, in South Korea, rural areas generally have a higher drinking rate with higher average ages [51], thus the alcohol consumption hypothesis could be associated with a higher association between PM$_{2.5}$ and violence in rural areas than in urban areas. However, further studies are required to examine the hypotheses.

This study has several strengths. First, we focused on understudied health outcomes: injuries from violence, which also include nonfatal cases that have been less examined in comparison to homicide. In addition, we used a nationwide dataset of daily modeled PM$_{2.5}$ concentrations and injuries from violence in South Korea to gain a more comprehensive understanding of the associations between PM$_{2.5}$ and violence across the country. The nationwide study could improve the generalizability and robustness of risk estimates compared to previous studies.

Nevertheless, this study has some limitations. First, this study used district-level residential addresses to assign exposures to air pollution, which may lead to measurement errors in exposure. However, these errors are likely to be random and the estimates of the effect can go towards the null [52]. Second, there could be some unmeasured time-varying confounders (i.e., high-risk drinking and daily mood) that might affect our estimates. Third, due to the limited data availability, we were unable to explore the associations between PM$_{2.5}$ and violence by individual socioeconomic factors, such as occupation or income levels, although these socioeconomic factors could be substantially associated with violence. Fourth, because our dataset only covered patients who were diagnosed admitted to, and discharged from the hospital, there was a limitation in capturing information on patients who were not admitted or who died before reaching the hospital. Also, the outcome definition relies on diagnosis codes, which may lead to an underestimation of the actual cases of violence. In addition, although we reported the descriptive statistics on subtypes of violence based on diagnostic code, we could not present the stratified association between PM$_{2.5}$ and violence by subtype because of the insufficient sample size of each subtype: the majority of violence cases were due to collisions (around 90%). It would be helpful to improve the understanding of the underlying mechanisms between air pollution and violence if future studies could examine the types of violence with larger prospective cohort data. Fifth, although the survey dataset aimed to get population representativeness through specific sampling methods, we acknowledge that there may still be potential bias, and caution is needed when interpreting the findings. Sixth, we could not perform two-pollutant models because of the data limitation regarding different modeled

pollutants (e.g., ozone and nitrogen dioxide). Lastly, our findings need to be further validated not only in South Korea but also in other regions and countries, especially those with different climate zones.

## Conclusions

Our study evaluated that short-term PM$_{2.5}$ exposures are associated with violence incidence in South Korea and found that the association was more prominent in males and younger populations than in females and older populations. Our study provides evidence for establishing more targeted action plans against PM$_{2.5}$ and violence at the national scale. In addition, our estimated excess violence burden attributable to non-compliance with WHO air quality guidelines and the NAAQS and KAS suggests the potential benefits of more stringent air quality standards aligned with global-standard air quality guidelines or standards.

## Supporting information

**S1 File. Supplementary materials.**
(DOCX)

## Author Contributions

**Conceptualization:** Jieun Oh, Whanhee Lee.

**Data curation:** Jiwoo Park, Ayoung Kim, Cinoo Kang, Dohoon Kwon, Jinah Park.

**Formal analysis:** Jiwoo Park, Jieun Oh, Hyewon Yoon.

**Funding acquisition:** Whanhee Lee.

**Methodology:** Whanhee Lee.

**Supervision:** Whanhee Lee.

**Visualization:** Jiwoo Park, Hyewon Yoon.

**Writing – original draft:** Jiwoo Park, Jieun Oh, Whanhee Lee.

**Writing – review & editing:** Ho Kim.

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
