## [Decision Letter · Decision Letter 0]

4 Oct 2024

PONE-D-24-37758Association between fine particulate matter (PM2.5) and violence incidence in South Korea: a nationwide time-stratified care-crossover studyPLOS ONE

Dear Dr. Lee,

Thank you for submitting your manuscript to PLOS ONE. After careful consideration, we feel that it has merit but does not fully meet PLOS ONE’s publication criteria as it currently stands. Therefore, we invite you to submit a revised version of the manuscript that addresses the points raised during the review process.

**ACADEMIC EDITOR: **Thank you for submitting your manuscript. The reviewers and I believe it is of potential value for our readers. However, the reviewers have raised a number of very important issues, and their excellent comments will need to be adequately addressed in a revision before the acceptability of your manuscript for publication in the Journal can be determined. We cannot guarantee that your revised paper will be chosen for publication; this would be solely based on how satisfactorily you have addressed the reviewer comments.

We look forward to receiving your revised manuscript.

Kind regards,

Dong Keon Yon, MD, FACAAI, FAAAAI

Academic Editor

PLOS ONE

Journal Requirements:

3. Your abstract cannot contain citations. Please only include citations in the body text of the manuscript, and ensure that they remain in ascending numerical order on first mention.

Additional Editor Comments:

Thank you for submitting your manuscript. The reviewers and I believe it is of potential value for our readers. However, the reviewers have raised a number of very important issues, and their excellent comments will need to be adequately addressed in a revision before the acceptability of your manuscript for publication in the Journal can be determined. We cannot guarantee that your revised paper will be chosen for publication; this would be solely based on how satisfactorily you have addressed the reviewer comments.

Reviewers' comments:

Reviewer's Responses to Questions

**Comments to the Author**

1. Is the manuscript technically sound, and do the data support the conclusions?

Reviewer #1: Partly

Reviewer #2: Partly

Reviewer #3: Yes

2. Has the statistical analysis been performed appropriately and rigorously? 

Reviewer #1: No

Reviewer #2: Yes

Reviewer #3: Yes

3. Have the authors made all data underlying the findings in their manuscript fully available?

Reviewer #1: No

Reviewer #2: Yes

Reviewer #3: Yes

4. Is the manuscript presented in an intelligible fashion and written in standard English?

Reviewer #1: Yes

Reviewer #2: Yes

Reviewer #3: Yes

5. Review Comments to the Author

Reviewer #1: Park et al. evaluated the association of PM2.5 with violence using a time-stratified case-crossover design and a conditional logistic regression. The manuscript is confusing and needs some modifications as follows:

(1) In many sections, the authors said that they predicted PM2.5, and it is not clear whether they did some imputations or just used the predicted values.

(2) There are several over-interpretations. This study found some associations between short-term PM2.5 exposure and the risk of violence. Some sentences, such as "6.21% of the excess violence was due to non-compliance with the WHO guidelines (daily PM2.5 > 15 µg/m3)" and "Our generalizable findings", seems to be over-interpreted.

(3) There are too less information in the Figure and Table legends. At least, what variables were used for the adjustments, what methods were used, and the acronyms need to be explained. In addition, what results are adjusted for daily mean temperature?

(4) Line 225, In particular, Particularly,

(5) "Estimation of the excess violence burden attributable to PM2.5" is highly confusing. "We calculated daily attributable estimates, and the sum of daily attributable violence cases represents the total excess violence cases". The authors have calculated daily average of violence cases, and when the number of cases is above the average, then it is considered as the excess violence cases. What is the purpose of these results? I think it only causes confusing.

(6) No interaction tests were done and there are some sentences about more prominent association of PM2.5 with violence incidence in males and younger populations.

Reviewer #2: Thank you for the opportunity to review this manuscript. This study aims to evaluate the association between fine particulate matter (PM2.5) and violence incidence in South Korea using a case-crossover design. While I find the study's findings significant, there are several points where additional explanations and analyses could strengthen the results.

My major concerns are as follows:

There is considerable overlap and redundancy between the information provided in the supplementary files and the main text. Particularly, details related to the Data Source, Outcome Definition, Study Design, Statistical Analysis, and Estimation of the Excess Violence are spread across both sections. I recommend including these key elements more clearly in the main text.

1. Lines 130-137: The authors should explain why they chose lag0-2 for their analysis.

2. Lines 162-164: The explanation provided here is insufficient. Could the authors expand on this section?

3. Lines 139-142: Could the authors consider further stratifying the age groups, similar to what was presented in Table 1 (e.g., 0-19, 20-39, 40-64, 65-79, 80+)? I believe there may be differences across age groups and occupations.

4. Table 1: I recommend presenting the cases by year and region.

5. How did the authors account for regional variations in their statistical analyses? I believe there could be differences based on geographical regions. If possible, I suggest conducting stratified analyses based on individual or regional income levels, followed by meta-analyses by city as part of a sensitivity analysis.

6. I am also curious if the authors can consider subtypes of violence. If possible, I recommend adding an analysis by types of violence. Furthermore, while the supplementary file includes some details, the authors should specify the ICD codes used for the classification of violence cases.

7. Table 2 and Figure 3: The authors claim that the excess burden for those aged 0-64 is higher than that for those aged 65 and above, but this varies considerably by season. For instance, in the cold season, the odds ratio and excess burden for the 65+ age group are much higher. I would be interested to know the authors’ thoughts on this discrepancy.

8 .Figure 2: The exposure-response curve suggests that concentrations below approximately 10 may not be associated with increased risk, indicating a potential threshold. What do the authors think about this? Given that linearity is observed at higher concentrations, I believe an additional analysis, such as piecewise regression considering a threshold, would be reasonable.

9. The authors emphasize WHO guidelines and NAAQS air quality standards. I recommend providing a more detailed explanation of these guidelines and discussing South Korea’s air pollution levels in the background section.

10. Lines 214-221, 302-305: The data used in this study is based on a sample of patients discharged from general hospitals. I believe this limitation should be considered when generalizing the study’s findings.

11. The outcome definition relies on diagnosis codes, which may lead to an underestimation of actual violence incidence. The authors should consider this potential limitation.

12. Lines 245-245: It is unclear whether PM2.5 exposure increases the likelihood of aggressive behavior or the risk of being harmed. "Violence incidence" seems to refer to those harmed by violence, but the authors should clarify this distinction. Additionally, as social and environmental factors may play a larger role in violent events, it would be helpful to address these aspects in the discussion.

Minor comments:

1. Abstract, Line 34: I believe there is a misstatement regarding the estimated excess fraction of violence cases attributable to PM2.5. The confidence intervals and estimates are unclear.

2. I recommend that the authors provide a table of contents and page numbers for the supplementary file.

3. The authors should link specific paragraphs in the supplementary file to the appropriate text in the manuscript (e.g., "2. Air Pollution Prediction Models" should reference "Supplementary Text 2").

4. Some results are missing odds ratios and 95% confidence intervals. I recommend addressing this (e.g., Lines 190-195).

5. Lines 211-212: This sentence could be moved to the Statistical Analysis or Sensitivity Analysis section.

6. Although Figure 1 provides PM2.5 levels, I recommend including the mean concentration of PM2.5 in the main text.

Reviewer #3: 1. In Table 1, the age is categorized as 0-19, 20-39, 40-64, 65-79, 80+, but I wonder if the actual analysis uses 0-64, 65+. I recommend analyzing the age according to Table 1. If you think that the analysis results by subdividing the age categories are not better than the existing analysis results, please express the age as under 65 and over 65 in Table 1.

2. There are treatment results in the data. Please classify them as follows and calculate the risk of no hope and death.

- Unknown means missing.

- No hope and death are one category.

- Remaining results are one category.

6. PLOS authors have the option to publish the peer review history of their article (what does this mean?). If published, this will include your full peer review and any attached files.

Reviewer #1: **Yes: **Seogsong Jeong

Reviewer #2: No

Reviewer #3: No

---

## [Author Response · Author response to Decision Letter 0]

5 Nov 2024

Dear Editor,

We sincerely thank the reviewers for their insightful comments on our manuscript. We have carefully revised the manuscript and responded to each comment point by point below.

Reviewer #1: Park et al. evaluated the association of PM2.5 with violence using a time-stratified case-crossover design and a conditional logistic regression. The manuscript is confusing and needs some modifications as follows:

#1.1. In many sections, the authors said that they predicted PM2.5, and it is not clear whether they did some imputations or just used the predicted values.

[Response] Thank you for your comment and our apologies for the confusion. In this study, we used “predicted values of PM2.5 concentrations”, which was driven by a machine-learning ensemble prediction model to cover unmonitored areas. In other words, this model predicted daily PM2.5 concentrations for both monitored and unmonitored areas during the study period, recent studies have adopted the modeled air pollution concentrations to provide nationwide risk estimates.1-3 To clarify it, we have revised the related text (lines 123-133 on page 7) and have included detailed explanations and performance for the PM2.5 prediction model in the Supplemental Materials “1. Air pollution prediction models” and Table S1. 

(Lines 123-130 on Page 7)

To cover unmonitored districts, a nationwide daily modeled PM2.5 (the predicted value using a machine-learning ensemble prediction model with a 1 km2 spatial resolution) was used as an exposure for all districts. The modeled PM2.5 (24-hour average) was provided by the AiMS-CREATE team, and their prediction models were used in previous studies.18,19 The ensemble model incorporates three machine-learning algorithms (random forest, extreme gradient boosting, and deep neural network). Detailed information on the model is reported in the Supplementary Materials, “1. Air pollution prediction models” and Supplementary Table 1 (Table S1).

#1.2. There are several over-interpretations. This study found some associations between short-term PM2.5 exposure and the risk of violence. Some sentences, such as "6.21% of the excess violence was due to non-compliance with the WHO guidelines (daily PM2.5 > 15 µg/m3)" and "Our generalizable findings", seems to be over-interpreted.

[Response] We agree with your comments and acknowledge that there is a possibility of misinterpretation. Thus, we have revised related sentences across the manuscript (main change: we used “related or associated” instead of “attributable” to avoid misinterpretation). We would be glad if you could consider our modifications below:

(Lines 267-268 on Page 14)

Approximately 6.21% (1.59–10.33) and 1.07% (0.28–1.78) of excess violence cases were related to non-compliance with the WHO air quality guidelines (15 µg/m3) and NAAQS (35 µg/m3), respectively.

(Lines 288-289 on Page 20)

We also found that the excess fractions of violence associated with non-compliance with WHO guidelines and NAAQS and KAS were 6.21% and 1.07%, respectively.

(Lines 34-37 on Page 2; Abstract)

The estimated excess fraction of violence cases related to PM2.5 was 14.53% (95% CI: 4.54%–22.92%), and 6.42% (95% CI: 1.97%–10.26%) of the excess violence was related to non-compliance with the WHO guidelines (daily PM2.5 > 15 µg/m3). Our findings might be the evidence for the need of establishing elaborate action plans and stricter air quality guidelines to reduce the hazardous impacts of PM2.5 on violence in South Korea.

#1.3. There are too less information in the Figure and Table legends. At least, what variables were used for the adjustments, what methods were used, and the acronyms need to be explained. In addition, what results are adjusted for daily mean temperature?

[Response] Our apologies for the lack of information on figures and tables. We have added detailed information to all figures and tables as below:

Figure 2. Flexible association between short-term exposure to PM2.5 and violence cases. Each Blue and Green dashed lines indicate WHO: World Health Organization air quality guidelines (daily average PM2.5: 15 µg/m3), NAAQS: National Ambient Air Quality Standards (daily average PM2.5: 35 µg/m3) in the United States, and KAS: Korean Air Quality Standards (daily average PM2.5: 35 µg/m3), respectively. A conditional logistic regression model, adjusted for holiday and daily mean temperatures, within a time-stratified case-crossover design was used to estimate odds ratios and 95% confidence intervals. We classified the study districts (si/gun/gu) into urban (si and gu) and rural (gun) districts. GRDP High and Low areas are separated by a median value.

Figure 3. Associations between short-term exposure to PM2.5 (lag 0–2) and violence cases by season and subgroup. OR: Odds ratio per 10 µg/m3 of PM2.5. A conditional logistic regression model, adjusted for holiday and daily mean temperatures, within a time-stratified case-crossover design was used to estimate odds ratios and 95% confidence intervals. Wald type test was used to test the effect modification. We classified the study districts (si/gun/gu) into urban (si and gu) and rural (gun) districts. GRDP High and Low areas are separated by a median value.

Table 2. Excess numbers and fractions of violence cases attributable to PM2.5 and non-compliance with the current WHO air quality guidelines and NAAQS in the United States. WHO: World Health Organization, NAAQS: National Ambient Air Quality Standards. KAS: Korean Air Quality Standards. We classified the study districts (si/gun/gu) into urban (si and gu) and rural (gun) districts. GRDP High and Low areas are separated by a median value.

In addition, yes, we adjusted daily mean temperatures as a confounder. The daily average temperatures are generally considered as a confounder, which could affect both the daily mean PM2.5 (our main exposure) and the daily violence counts (outcome),4,5 when examining the association between air pollutants and health outcomes. Therefore, for consistency, we also controlled for daily mean temperatures in this study as previous studies performed.6,7

#1.4. Line 225, In particular, Particularly,

[Response] Thank you for pointing it out. We have revised the text. 

#1.5. "Estimation of the excess violence burden attributable to PM2.5" is highly confusing. "We calculated daily attributable estimates, and the sum of daily attributable violence cases represents the total excess violence cases". The authors have calculated daily average of violence cases, and when the number of cases is above the average, then it is considered as the excess violence cases. What is the purpose of these results? I think it only causes confusing.

[Response] We appreciate your careful comments. First, we used the daily sum of violence cases (counts) and calculated excess violence counts related to daily PM2.5 using the corresponding odd ratio (OR) for the daily PM2.5.8,9 Through this procedure, we ultimately aimed to quantify and present “the excess burden” of violence related to “the whole range of PM2.5 during the entire study period” (i.e., the actual/quantitative impact of the exposure) because OR could provide only the risk ratio per unit increase of PM2.5. This information is critical for the planning and evaluation of public health interventions, and it is better provided by relative excess measures such as the attributable fraction (AF), or absolute excess measures such as the attributable number (AN).10 To clarify it, we have revised the related texts with the necessity of the excess measures and detailed calculation procedures. Please see below:

(Lines 194-210 on Pages 10-11)

Excess violence burden related to PM2.5

Odd ratio estimates for short-term exposure to PM2.5 were translated into excess violence cases related to PM2.5, to demonstrate the excess burden due to PM2.5 exposures during the study period. The estimates of the excess burden (either as an attributable burden or relative excess measures of outcomes) are widely used to assess the actual and quantitative public health impacts of extreme exposures (e.g., heatwaves or high-level air pollution). 

To calculate it, we created district-specific time-series datasets including daily mean PM2.5 and daily counts of violence. For each time-series data, we calculated the daily excess violence cases attributable to PM2.5 using the corresponding estimated ORs for the PM2.5 concentrations of each day (daily excess violence cases related to daily PM2.5: 〖sum of violecne cases〗_t *[〖OR〗_t-1]/〖OR〗_t; where t indicates the day, and ORt presents the daily OR for the PM2.5 concentrations of day t). The sum of daily excess violence cases indicates the total excess violence cases related to PM2.5 during the study period, and its ratio with the total number of violence provides the total excess fraction (%) of violence cases related to short-term PM2.5 exposures. 

#1.6. No interaction tests were done and there are some sentences about more prominent association of PM2.5 with violence incidence in males and younger populations.

[Response] We sincerely appreciate your comment. In accordance with your suggestion, we have tested the effect modification of the association between PM2.5 and violence cases using the Wald test, and the results have been added to the revised Figure 3. Further, we described the statistically meaningful results in the main text. 

(Lines 255-262 on Pages 13-14)

Figure 3 shows the association between PM2.5 and violence in the total population and by sub-group. In the total population, the association between PM2.5 and violence was evident with an OR of 1.07 with 95% CI: 1.02–1.12. The association was weakly higher in rural areas (OR: 1.23, 95% CI: 1.02–1.48) than in urban areas (OR: 1.06, 95% CI: 0.99–1.12) (Wald test p-value: 0.096). Further, the association between PM2.5 and violence was statistically pronounced in cold seasons (OR: 1.08, 95% CI: 1.02–1.14) compared to warm seasons (1.05, 0.96–1.15), and the association was more pronounced in people aged 65 (OR:1.33, 95% CI: 1.04–1.71) or older than those aged 64 or less in cold seasons (Wald test p-value: 0.042).

(Revised) Figure 3. Associations between short-term exposure to PM2.5 (lag 0–2) and violence cases by season and subgroup. OR: Odds ratio per 10 µg/m3 of PM2.5. A conditional logistic regression model, adjusted for holiday and daily mean temperatures, within a time-stratified case-crossover design was used to estimate odds ratios and 95% confidence intervals. Wald type test was used to test the effect modification. We classified the study districts (si/gun/gu) into urban (si and gu) and rural (gun) districts. GRDP High and Low areas are separated by a median value.

Reviewer #2: Thank you for the opportunity to review this manuscript. This study aims to evaluate the association between fine particulate matter (PM2.5) and violence incidence in South Korea using a case-crossover design. While I find the study's findings significant, there are several points where additional explanations and analyses could strengthen the results.

My major concerns are as follows:

There is considerable overlap and redundancy between the information provided in the supplementary files and the main text. Particularly, details related to the Data Source, Outcome Definition, Study Design, Statistical Analysis, and Estimation of the Excess Violence are spread across both sections. I recommend including these key elements more clearly in the main text.

[Response] Thank you for your careful comments. Following your suggestion, we have tried to re-organize our manuscripts to avoid overlap and redundancy. Please see our revised files. We truly appreciate your opinion.

#2.1. Lines 130-137: The authors should explain why they chose lag0-2 for their analysis.

[Response] We are thankful for your feedback. Since violence cases usually occur as acute events, we adopted two days as a lag period based on previous relevant studies 11-13. We also thought the selection could increase the comparability with existing findings. We added more rationales and explanations for the lag period in the methods section like below:

(Lines 164-166 on Page 9)

For the main model, we selected a mean value of lag 0 to lag 2 PM2.5 exposure to address the average health risks associated with the same and the previous days’ exposures based on existing relevant studies.21,31,32

(Lines 171-173 on Page 9)

The relationship between relatively short-term temperature (one or two lag days) and violence, suicide, or other acute mental disorders has been identified in many related epidemiological studies.13-16

Additionally, we have performed sensitivity analyses regarding different lag days to check the robustness of our results, and the results have been added to Table S8. Our main results were consistent with different lag periods. 

#2.2. Lines 162-164: The explanation provided here is insufficient. Could the authors expand on this section?

[Response] We greatly appreciate your comment. We have added more detailed explanations for the sensitivity analyses in the methods section below:

(Lines 226-232 on Page 12)

We performed a series of sensitivity analyses to examine whether our results were consistent with additional different modeling specifications in the total population. Sensitivity analyses were carried out about two aspects. First, we tried to show the robustness of PM2.5’s lag period. Single lag from 0 to 3 was executed and the moving average from lag 0-1 to 0-3 was tested. Second, for daily temperature, which was used as a covariate, we applied 3 and 7 days of lag period, and 3 and 5 degrees of freedom for an exposure-response relationship.

#2.3. Lines 139-142: Could the authors consider further stratifying the age groups, similar to what was presented in Table 1 (e.g., 0-19, 20-39, 40-64, 65-79, 80+)? I believe there may be differences across age groups and occupations.

[Response] We would like to express our gratitude for your meaningful comment. As you pointed out, we have added more specific results for stratifying age groups in Table S7. We found that the findings were consistent with the main results; i.e., a statistically significant association was found in the younger age group (20–39 years) rather than the older age groups (65–79 years and 80+ years) in total seasons, and the risk in the older age groups (65–79 years and 80+ years) was greater in the cold seasons than in the warm seasons. Therefore, we have added a sentence in the result section as follows:

(Lines 262-263 on Page 14)

This pattern remained consistent even when age groups were further subdivided (Table S7).

Table S7. Associations between short-term exposure to PM2.5 (lag 0–2) and violence cases by season and age group.

Seasons Age group OR (95% CI)

Every season Total 1.07 (1.02, 1.12)

 0-19 years 1.09 (0.96, 1.24)

 20-39 years 1.11 (1.03, 1.20)

 40-64 years 1.02 (0.95, 1.11)

 65-79 years 1.11 (0.89, 1.39)

 80+ years 1.75 (0.96, 3.19)

Warm seasons Total 1.05 (0.96, 1.15)

 0-19 years 1.04 (0.83, 1.30)

 20-39 years 1.18 (1.01, 1.37)

 40-64 years 0.99 (0.86, 1.14)

 65-79 years 0.87 (0.59, 1.28)

 80+ years 0.84 (0.29, 2.46)

Cold seasons Total 1.08 (1.02, 1.14)

 0-19 years 1.09 (0.94, 1.28)

 20-39 years 1.09 (0.99, 1.20)

 40-64 years 1.03 (0.94, 1.13)

 65-79 years 1.30 (0.98, 1.72)

　 80+ years 2.52 (1.04, 6.12)

However, due to the limited data availability (the surveillance system database we used in this study does not include the information on enrollees’ occupation and income status), we were unable to consider an individual's occupation, and we have addressed this issue as a major limitation in the discussion section:

(Lines 378-381 on Page 24)

Third, due to the limited data availability, we were unable to explore the associations between PM2.5 and violence by individual socioeconomic factors, such as occupation or income levels, although these socioeconomic factors could be substantially associated with violence.

#2.4. Table 1: I recommend presenting the cases by year and region.

[Response] We are grateful for your comment. We have included the descriptive information about regions using the urban/rural indicator and gross regional domestic product (GRDP) per person in the revised Table 1. Ad

---

## [Decision Letter · Decision Letter 1]

26 Nov 2024

PONE-D-24-37758R1Association between fine particulate matter (PM2.5) and violence cases in South Korea: a nationwide time-stratified care-crossover studyPLOS ONE

Dear Dr. Lee,

Thank you for submitting your manuscript to PLOS ONE. After careful consideration, we feel that it has merit but does not fully meet PLOS ONE’s publication criteria as it currently stands. Therefore, we invite you to submit a revised version of the manuscript that addresses the points raised during the review process.

Please see the minor comments.

We look forward to receiving your revised manuscript.

Kind regards,

Johanna Pruller,

Associate Editor

PLOS ONE

on behalf of

Dong Keon Yon, MD, FACAAI, FAAAAI

Academic Editor

PLOS ONE

Journal Requirements:

Additional Editor Comments:

Please see the minor comments.

Reviewers' comments:

Reviewer's Responses to Questions

**Comments to the Author**

1. If the authors have adequately addressed your comments raised in a previous round of review and you feel that this manuscript is now acceptable for publication, you may indicate that here to bypass the “Comments to the Author” section, enter your conflict of interest statement in the “Confidential to Editor” section, and submit your "Accept" recommendation.

Reviewer #1: All comments have been addressed

Reviewer #2: All comments have been addressed

Reviewer #3: All comments have been addressed

2. Is the manuscript technically sound, and do the data support the conclusions?

Reviewer #1: Yes

Reviewer #2: Yes

Reviewer #3: Yes

3. Has the statistical analysis been performed appropriately and rigorously? 

Reviewer #1: I Don't Know

Reviewer #2: Yes

Reviewer #3: Yes

4. Have the authors made all data underlying the findings in their manuscript fully available?

Reviewer #1: No

Reviewer #2: Yes

Reviewer #3: Yes

5. Is the manuscript presented in an intelligible fashion and written in standard English?

Reviewer #1: Yes

Reviewer #2: Yes

Reviewer #3: Yes

6. Review Comments to the Author

Reviewer #1: The authors have addressed most of my concerns.

Although there is a possibility of misinterpretation for “excessive burden due to PM2.5” but I understand that the method is frequently used.

Reviewer #2: I would like to thank the authors for their thoughtful responses to my concerns.

I have a few additional minor comments, and I would appreciate it if these could be considered.

[1] Line 95: "We obtained national data on hospital visits due to violence from 2015 to 2019 across247"

I suggest correcting "across247" to "across 247".

[2] Line 115: "Each recorded case also includes information on the patient’s sex, age, residential address (ZIP code), date of hospitalization"

While the residential address in South Korea can be considered similar to a ZIP code, it is not identical. I recommend modifying this expression to reflect that they are not exactly the same.

[3] Regarding Excess violence, typically, excess mortality or excess hospitalization is calculated using relative risk (RR). In cases where the disease incidence is low (rare disease), OR approximates RR, so using OR as a substitute for RR is not a significant concern. However, the authors should consider adding a brief explanation in the text to clarify this relationship between OR and RR.

Reviewer #3: (No Response)

7. PLOS authors have the option to publish the peer review history of their article (what does this mean?). If published, this will include your full peer review and any attached files.

Reviewer #1: No

Reviewer #2: **Yes: **Jongmin Oh

Reviewer #3: No

---

## [Author Response · Author response to Decision Letter 1]

26 Nov 2024

Dear Editor,

We sincerely thank the reviewers for their insightful comments on our manuscript. We have carefully revised the manuscript and responded to each comment point by point below.

Journal Requirements:

[Response] We truly appreciate your careful check and would like to express our apologies. We could not find that there was an error in the Endnote we used, and thus previous reference list was incorrect. Once again, we apologize for the confusion. In accordance with the criteria you mentioned, we have revised and double-checked references, and we found that the current references in the revised manuscript are correct. Please the modifications we performed below:

(List of the reference modifications)

#1. “14. Burkhardt J, Bayham J, Wilson A, et al. The relationship between monthly air pollution and violent crime across the United States. Journal of Environmental Economics and Policy 2020; 9(2): 188-205”. This study includes all countries of the United States. Thus, this is not a suitable reference to our description of “many existing studies were conducted in the restricted areas”. We replaced this reference as the “Nguyen A-M, Malig BJ, Basu R. The association between ozone and fine particles and mental health-related emergency department visits in California, 2005–2013. PloS one 2021; 16(4): e0249675.”, which is the study conducted in California with monitored air pollutants. 

#2. We deleted previous References 27-30 because they have similar information (why the time-stratified case-crossover design is suitable to evaluate the impacts of short-term exposure to environmental stressors and health outcomes), compared to previous References 21-26.

#3. (Lines 174). We found the error in the reference number: the previous manuscript had a “13-16” reference, which is incorrect. We have revised it with corrected references. 

#4. (Lines 217) We found the error in the reference number. The previous manuscript had a “12, 17-20” reference, which is incorrect. We have revised it with a corrected reference showing the importance of the health impacts of low PM2.5 exposures.

#5. (Lines 232) We also found that there was an error in the reference number regarding the Monte Carlo simulation. We have revised it with the related references.

#6. (Line 110) We have added the references regarding the Neyman allocation method. 

Reponses to Reviewers

Reviewer #1: The authors have addressed most of my concerns.

#1.1. Although there is a possibility of misinterpretation for “excessive burden due to PM2.5” but I understand that the method is frequently used.

[Response] We truly appreciate your understanding. As you mentioned, we recognized that the expression “excess burden due to” has been widely used in the relevant studies 1-4, even though this word has a possibility of misinterpretation related to causality or conceptual confusion between excess risk (RR1-RR2) and attributable risk ([RR1-RR2]/RR1). We tried to use the attributable risk as a measure of the proportion of the outcome occurrence that can be attributed to ambient PM2.5 exposures, thus this definition can be expressed by estimating excess risk as [RR1-RR2]/RR1. This formula provides the proportion of the excess risk of outcome that can be attributed to the exposure.5 

Thus, we cautiously thought that this concept was aligned with the measurement we would like to estimate. Nonetheless, to avoid potential confusion and clarify the concept we adopted, we have revised all related terms to “excess burden/number attributable to PM2.5” across the manuscript. Once again, we are grateful for your generous understanding. 

Reviewer #2: I would like to thank the authors for their thoughtful responses to my concerns.

I have a few additional minor comments, and I would appreciate it if these could be considered.

#2.1. Line 95: "We obtained national data on hospital visits due to violence from 2015 to 2019 across247" I suggest correcting "across247" to "across 247".

[Response] Thank you so much for your careful check. We have revised it. 

#2.2. Line 115: "Each recorded case also includes information on the patient’s sex, age, residential address (ZIP code), date of hospitalization". While the residential address in South Korea can be considered similar to a ZIP code, it is not identical. I recommend modifying this expression to reflect that they are not exactly the same.

[Response] Thank you for pointing out the important point. It was our mistake. It was not a Zip code. It should be the district called “si-gun-gu” in Korean. We sincerely apologize for our mistake. Please see our modification below:

(Lines 115-116 on Page 7)

Each recorded case also includes information on the patient’s sex, age, residential address (districts, “si/gun/gu” in Korean),

#2.3. Regarding Excess violence, typically, excess mortality or excess hospitalization is calculated using relative risk (RR). In cases where the disease incidence is low (rare disease), OR approximates RR, so using OR as a substitute for RR is not a significant concern. However, the authors should consider adding a brief explanation in the text to clarify this relationship between OR and RR.

[Response] We genuinely appreciate your suggestion. Yes, as you mentioned, the odd ratio (OR) can be regarded as an approximated estimate of the relative risk (RR) when the outcome risk is low in both groups (in general, 20% or less).6 The total number of outcomes (violence cases) was 2,867 cases during the find years (2015 to 2019), and when the total number of investigation cases of the Korea National Hospital Discharge In-depth Injury Survey (sampled 150,000 to 300,000; see lines 102), the probability of our outcome was substantially small. Thus, as you said, we also thought that our OR estimates could be good approximated values of RRs that are needed to calculate the attributable fraction. In accordance with your suggestion, we have added the related text in the Method section. 

(Lines 204-212 on Page 11)

Herein, the OR estimates were used as the approximated values of relative risk (RR) that were originally used to calculate the attributable risk estimates. When the proportion of the outcome onset is substantially small (in general, 20% or less), the OR estimates can be regarded as an approximated estimate of the RR estimates 35. In this study, the total number of outcomes (violence cases) was 2,867 cases during the find years (2015 to 2019), and when the total number of investigation cases of the Korea National Hospital Discharge In-depth Injury Survey (sampled 150,000 to 300,000; see lines 101). Therefore, the probability of our outcome was substantially small. Thus, we used OR estimates as the approximated values of RR estimates in this study.

References for the responses

1. Gasparrini A, Guo Y, Hashizume M, et al. Mortality risk attributable to high and low ambient temperature: a multicountry observational study. The Lancet 2015; 386(9991): 369-75.

2. Lee W, Prifti K, Kim H, et al. Short-term Exposure to Air Pollution and Attributable Risk of Kidney Diseases: A Nationwide Time-series Study. Epidemiology 2022; 33(1).

3. Min J, Kang D-H, Kang C, et al. Fluctuating risk of acute kidney injury-related mortality for four weeks after exposure to air pollution: A multi-country time-series study in 6 countries. Environment International 2024; 183: 108367.

4. Weinberger KR, Harris D, Spangler KR, Zanobetti A, Wellenius GA. Estimating the number of excess deaths attributable to heat in 297 United States counties. Environmental Epidemiology 2020; 4(3).

5. Thelle DS, Laake P. Chapter 9 - Epidemiology. In: Laake P, Benestad HB, Olsen BR, eds. Research in Medical and Biological Sciences (Second Edition). Amsterdam: Academic Press; 2015: 275-320.

6. Cook A, Sheikh A. Descriptive statistics (Part 2): Interpreting study results. Primary Care Respiratory Journal 2000; 8(1): 16-7.

---

## [Editor Report · Decision Letter 2]

3 Dec 2024

Association between fine particulate matter (PM2.5) and violence cases in South Korea: a nationwide time-stratified care-crossover study

PONE-D-24-37758R2

Dear Dr. Lee,

We’re pleased to inform you that your manuscript has been judged scientifically suitable for publication and will be formally accepted for publication once it meets all outstanding technical requirements.

Kind regards,

Dong Keon Yon, MD, FACAAI, FAAAAI

Academic Editor

PLOS ONE

Additional Editor Comments (optional):

This is an excellent paper.
---

## [Editor Report · Acceptance letter]

6 Dec 2024

PONE-D-24-37758R2 

PLOS ONE

Dear Dr. Lee, 

I'm pleased to inform you that your manuscript has been deemed suitable for publication in PLOS ONE. Congratulations! Your manuscript is now being handed over to our production team.

Kind regards, 

on behalf of

Dr. Dong Keon Yon 

Academic Editor

PLOS ONE